

# DDSUD: dynamically detecting subsequence uncertainty and diversity for active learning in imbalanced Chinese sentiment analysis

Shufeng Xiong[1], Yibo Si[1], Guipei Zhang[1], Bingkun Wang[2], Guang Zheng[1] and Haiping Si[1]

[1] College of Information and Management Science, Henan Agricultural University, Zhengzhou, China
[2] School of Information Engineering, Zhengzhou College of Finance and Economics, Zhengzhou, China

## ABSTRACT

Sentiment structure analysis in Chinese text typically relies on supervised deep-learning methods for sequence labeling. However, obtaining large-scale labeled datasets is both resource-intensive and time-consuming. To address these challenges, this study proposes Dynamically Detecting Subsequence Uncertainty and Diversity (DDSUD), a Bidirectional Encoder Representations from Transformers (BERT)-based active learning framework designed to tackle subsequence uncertainty and enhance the diversity of imbalanced datasets. DDSUD combines subsequence uncertainty detection, diversity-driven sample selection, and dynamic weighting, enabling an adaptive balance between these factors throughout the active learning iterations. Experimental results show that DDSUD achieves performance close to fully supervised training schemes with only 50% of the data labeled, and outperforms other state-of-the-art active learning methods with the same amount of labeled data. Moreover, by dynamically adjusting the trade-off between subsequence uncertainty and diversity, DDSUD demonstrates strong adaptability and generalization capability in low-resource environments, especially in handling imbalanced datasets, significantly improving the recognition of minority class samples.

Corresponding authors
Bingkun Wang,
wangbingkun@zzife.edu.cn
Haiping Si, haiping@henau.edu.cn

## INTRODUCTION

Sentiment classification plays a crucial role in natural language processing (NLP) with broad applications in social media monitoring, market research, and customer feedback analysis (*Kamal & Himel, 2023*; *Rane et al., 2024*). By identifying the emotional tone of a text—positive, negative, or neutral—sentiment analysis enables organizations to extract actionable insights from large-scale data. It facilitates public opinion tracking, product improvement, and strategic decision-making (*Yaqub et al., 2021*; *Birjali, Kasri & Beni-Hssane, 2021*).

Traditional sentiment classification methods rely on machine learning algorithms such as Support Vector Machines (SVMs), Convolutional Neural Networks (CNNs), and

feature-based models. While effective to some extent, these approaches depend heavily on handcrafted features and are limited in handling large, unstructured data (*Mumuni & Mumuni, 2024*). Deep learning methods like CNNs and Recurrent Neural Networks (RNNs) improve feature representation through automatic learning (*Hu et al., 2020*), but they require extensive labeled data, which is costly and time-consuming to obtain (*Albahri et al., 2023*). Furthermore, these models often perform poorly on imbalanced datasets, where minority sentiments such as negative emotions are underrepresented (*Lu, Ehwerhemuepha & Rakovski, 2022*).

Active learning offers a compelling solution by enabling models to selectively query the most informative samples for annotation, thereby improving performance with fewer labeled examples (*Ren et al., 2021*; *Fang, Li & Cohn, 2017*). This approach is particularly beneficial in sentiment analysis, where manual annotation is expensive. Moreover, active learning can mitigate class imbalance by emphasizing underrepresented classes during sample selection (*Schröder & Niekler, 2020*).

Despite these advantages, applying active learning to sentiment classification remains challenging. Traditional strategies such as uncertainty sampling prioritize low-confidence predictions (*Li et al., 2024*), which may not effectively capture subtle emotional cues, especially in short and context-dependent texts. Additionally, these methods often exacerbate class imbalance by favoring majority class examples, leading to biased models (*Lipton, 2018*; *Lango, 2019*). Most existing frameworks also lack adaptive mechanisms to adjust sampling strategies based on evolving data distributions (*Settles, 2011*).

These challenges are further amplified in Chinese sentiment structure analysis. Unlike English sentiment classification, which primarily involves polarity detection, Chinese tasks often require identifying fine-grained semantic roles such as emotional cause, target, and emotion type. These roles are frequently embedded in complex syntactic constructions with blurred boundaries and long-distance dependencies, increasing annotation and modeling difficulty. Moreover, Chinese social media texts exhibit flexible syntax, frequent subject omission, and non-standard expressions, demanding strong contextual reasoning. In contrast, English texts generally have more stable syntactic structures, making existing models more transferable.

Although pretrained models such as Bidirectional Encoder Representations from Transformers (BERT) and Robustly Optimized BERT Pretraining Approach (RoBERTa) are available for Chinese, they typically inherit training objectives and data characteristics from English-centric corpora, limiting their effectiveness in modeling implicit sentiment relationships in Chinese. Our experiments also show that token-level uncertainty estimation—commonly used in English active learning is inadequate for capturing the semantic span of sentiment-bearing expressions in Chinese. These limitations often result in low-quality sampling decisions and incomplete data coverage.

To address these issues, we propose Dynamically Detecting Subsequence Uncertainty and Diversity (DDSUD), a novel BERT-based active learning framework specifically designed for imbalanced Chinese sentiment classification. DDSUD introduces three key components: (1) subsequence uncertainty detection to capture fine-grained ambiguity; (2) diversity-driven sample selection to ensure representational richness; and (3) a

dynamic weighting mechanism to balance uncertainty and diversity throughout the active learning process.

This article is structured as follows. "Related Work" provides a review of related work on sequence labeling, active learning, and class imbalance. "Methodology" introduces the DDSUD framework, detailing its core components and scoring mechanism. "Experiments and Results Analysis" presents the experimental setup, datasets, baseline methods, and performance evaluations. "Conclusion" concludes the study and outlines directions for future research.

The main contributions of this work are as follows:

- We propose DDSUD, a novel active learning strategy that systematically integrates subsequence uncertainty and diversity to improve performance on imbalanced Chinese sentiment tasks.
- We introduce a dynamic weighting mechanism that adaptively balances uncertainty and diversity across learning iterations.
- We conduct extensive experiments on a benchmark Chinese dataset, showing that DDSUD achieves comparable or superior performance to fully supervised models with significantly fewer labeled samples.

# RELATED WORK

## Sequence labeling methods in deep learning

In natural language processing, core sequence labeling tasks—including part-of-speech tagging, named entity recognition, semantic role labeling, and aspect extraction—constitute foundational analytical layers. By detecting grammatical structures, predicate-argument relationships, and domain-specific elements (*e.g.*, sentiment targets), these tasks provide critical feature representations for downstream applications such as sentiment analysis and slot filling systems. Sentiment analysis, which involves identifying and classifying opinions expressed in text, is an essential application in NLP, as it provides valuable insights into public opinion, social trends, and customer feedback (*Jim et al., 2024*).

The advent of deep learning has revolutionized traditional methods in NLP, shifting them from rule-based and statistical approaches to models defined by enhanced accuracy and contextual awareness. Deep learning techniques have brought a paradigm shift in sentiment analysis (*Sharma, Ali & Kabir, 2024*), enabling models to better capture contextual nuances and improve accuracy.

### Part-of-speech tagging

Early part-of-speech (POS) tagging techniques relied on rule-based systems that utilized manually crafted linguistic features (*Berger, Della Pietra & Della Pietra, 1996*). Statistical methods, including Hidden Markov Models (HMM) (*Lee, Tsujii & Rim, 2000*) and Maximum Entropy Models (MaxEnt) (*Baldwin, 2009*), improved performance by leveraging data-driven techniques. Machine learning models, such as SVMs and

Conditional Random Fields (CRFs), further enhanced generalization. Nevertheless, deep learning has set new benchmarks, with models such as Recurrent Neural Networks (RNNs), Long Short-Term Memory (LSTM) networks (*Hu, Hou & Liu, 2024*), and Transformers achieving considerable success in capturing long-range dependencies and contextual information (*Azmi et al., 2025*).

### Word segmentation

Word segmentation has progressed from rule-based to statistical methods, and more recently, to deep learning techniques. Traditional approaches encountered challenges due to linguistic ambiguities, whereas statistical models, such as HMM and MaxEnt, mitigated some issues but were constrained by the reliance on manually designed features. More recently, RNNs, LSTMs, and Transformers (*Vaswani et al., 2017*) have significantly advanced word segmentation by modeling complex language patterns. Pre-trained models, such as BERT, have further advanced the field, significantly enhancing segmentation accuracy (*Wei & Guo, 2024*).

### Deep learning architectures

Deep learning models, particularly BERT and transformer-based architectures, have significantly advanced sentiment analysis by capturing bidirectional context and improving accuracy (*Kokab, Asghar & Naz, 2022*), especially for complex tasks like sentiment classification. When paired with active learning, these models can effectively address challenges such as imbalanced datasets by prioritizing informative samples, reducing the reliance on large labeled datasets. Hybrid models, such as Bidirectional Encoder Representations from Transformers—Bidirectional Long Short-Term Memory—Conditional Random Field (BERT-BiLSTM-CRF) (*Zhang et al., 2019*), further enhance performance by integrating CRFs, which improve class balance and provide better representation of underrepresented sentiment classes (*Bai et al., 2025*). These architectures have become indispensable tools in both active learning and sentiment analysis, boosting model efficiency and classification performance, particularly in resource-constrained environments.

## Sequence labeling with active learning

Sentiment classification, especially when dealing with imbalanced datasets, often relies on extensive labeled datasets, the creation of which is both labor-intensive and time-consuming (*Wertz et al., 2023*). Active learning offers an effective solution to this challenge by prioritizing the annotation of highly informative samples, with a particular focus on those belonging to underrepresented sentiment classes. By selectively sampling these data points, active learning not only optimizes the utilization of labeled data but also enhances model performance by concentrating on instances that contribute most significantly to improving the predictive capabilities of the model (*Jain & Kapoor, 2009*). This targeted approach facilitates more efficient learning and addresses the limitations imposed by class imbalance.

### Uncertainty-based methods

Uncertainty-based methods, such as Least Confidence (LC) (*Agrawal, Tripathi & Vardhan, 2021*) and Token Entropy (TE) (*Jacobs et al., 2021*), are designed to select samples where the model exhibits maximum uncertainty. While these methods have proven effective in sequence labeling tasks, they may prioritize more complex sequences or fail to capture the full diversity of the dataset. Advanced strategies, such as Lowest Token Probability (LTP) (*Liu, Benjamin & Zador, 2025*) and Bayesian Active Learning by Disagreement (BALD) (*Weerasooriya, 2024*), address these limitations by incorporating token-level uncertainty or leveraging model disagreement.

### Diversity-based methods

Diversity-based active learning methods aim to improve generalization by selecting samples that differ significantly in the feature space. Classical techniques such as maximizing margin distances (*Bauer et al., 2020*) and maximizing class coverage (*Huang et al., 2016*) help reduce sample redundancy to some extent. However, these methods typically treat entire sentences as atomic units and focus only on inter-sentence differences, overlooking which specific parts of a sentence actually contain informative content.

This limitation is particularly evident in sentiment structure recognition tasks, where key emotional elements—such as causes or targets—often reside within short phrases or subsequences, while the rest of the sentence may carry limited relevance. Relying on global sentence-level representations to assess sample diversity risks ignoring these high-value regions, leading to suboptimal selection.

To address this, DDSUD departs from full-sentence modeling and instead segments each sentence into multiple subsequences that may contain sentiment-relevant components. By comparing differences among these subsequences, DDSUD identifies samples that diverge more significantly in key semantic positions. This enhances the precision of the selection process and improves the model's ability to distinguish structurally different samples that might appear similar at the sentence level.

Moreover, under severe class imbalance, this approach helps the model avoid overfitting to typical expressions of frequent classes, and instead prioritize structurally informative examples from low-resource categories. As a result, the model achieves better generalization while improving class-level discrimination.

### Hybrid methods

Hybrid active learning methods select training samples by simultaneously considering model uncertainty and inter-sample diversity. For instance, Batch Active learning by Diverse Gradient Embeddings (BADGE) (*Pelicon et al., 2024*) selects data points that the model is currently uncertain about and that differ significantly from previously selected samples, thereby improving both representativeness and learning efficiency. Compared to strategies that rely solely on uncertainty or diversity, these methods more effectively balance model blind spots and data coverage.

However, such methods typically operate at the sentence level and cannot determine which parts of a sentence actually contain the most valuable information (*Radmard, Fathullah & Lipani, 2021*). In addition, they often use fixed schemes to combine uncertainty and diversity, lacking the flexibility to adapt to changes in model status or data distribution during training. As a result, their performance is limited in tasks involving complex structure or severe class imbalance.

To overcome these limitations, DDSUD introduces a hybrid sample selection strategy at the subsequence level. During each selection step, DDSUD segments a sentence into multiple spans and evaluates which ones exhibit both high uncertainty and high diversity. This enables more accurate identification of informative training samples. Furthermore, DDSUD incorporates a dynamic weighting mechanism that adjusts the influence of uncertainty and diversity based on the model's evolving state, thereby improving adaptability and practical effectiveness in real-world training scenarios.

## Sample information

In active learning, informativeness typically refers to the extent to which a sample helps reduce the model's predictive uncertainty. Samples with higher informativeness are more likely to improve the model's decision-making ability. Existing approaches often evaluate informativeness at the sentence level by selecting samples with the highest prediction uncertainty for annotation (*Schmidhuber, 2015*), which has shown effectiveness in tasks such as text classification.

However, these sentence-level strategies operate at a coarse granularity and fail to capture local differences in information distribution within the sentence. In many real-world texts, critical information is not evenly distributed but instead concentrated in specific phrases or structural components. For example, in sentiment structure recognition tasks, key elements such as emotion expressions, causes, or targets are often confined to short subsequences. Sentence-level evaluation tends to obscure such fine-grained uncertainty, causing the model to overlook the most informative regions.

*Radmard, Fathullah & Lipani (2021)* proposed a subsequence-based uncertainty sampling method that identifies uncertain spans within sentences to guide data selection. While this approach improves local sensitivity, it does not incorporate inter-sample diversity and lacks a mechanism for dynamically adjusting selection criteria during training.

To overcome these limitations, DDSUD introduces a subsequence-level information modeling framework. During each sampling iteration, the model segments input sentences into multiple candidate spans, evaluates their uncertainty and representativeness, and selects samples that contain the most informative segments for annotation. This strategy enables the model to focus on information-dense regions and, combined with a dynamic weighting mechanism, enhances the adaptability and effectiveness of the sampling process—particularly in structurally complex or class-imbalanced scenarios.

### Sentiment structure analysis in Chinese text

Sentiment structure analysis plays a crucial role in NLP, particularly in tasks such as opinion mining and social media sentiment detection (*Soong et al., 2019*), where understanding nuanced emotional expressions is vital. The complexity of Chinese textual sentiment analysis has garnered significant attention in recent years, driven by challenges such as word meaning ambiguity, contextual variations, and intricate sentence structures, compounded by the scarcity of comprehensive annotated datasets. These challenges have motivated researchers to explore both foundational and advanced methods to overcome the limitations of existing approaches.

Early methods relied on rule-based systems and statistical models, such as HMMs and Maximum Entropy (MaxEnt) (*Malouf, 2010*). While these approaches leveraged labeled data to improve performance, they were constrained by their dependence on handcrafted features and limited capacity to model long-range dependencies.

The advent of deep learning has revolutionized the field, with models such as RNNs, LSTM networks, and transformer-based architectures addressing longstanding challenges by capturing contextual nuances and modeling complex relationships (*Airlangga, 2024*). Building on these foundations, fine-tuned pre-trained models like RoBERTa and A Lite BERT (ALBERT) (*Özkurt, 2024*) have significantly advanced sentiment classification by capturing bidirectional context and reducing reliance on extensive labeled datasets.

Despite the substantial improvements in sentiment classification performance brought by deep learning, the need for large amounts of labeled data remains a significant challenge, particularly given the scarcity of such data in practical applications. To address this, active learning techniques, such as Bayesian Active Learning by Disagreement (BALD) (*Huang et al., 2023*), prioritize annotating informative samples while ensuring balanced class representation.

## METHODOLOGY

### Overview of DDSUD

This article proposes a BERT-based active learning method, named Dynamically Detecting Subsequence Uncertainty and Diversity (DDSUD), which overcomes the limitations of traditional methods that rely solely on sentence-level or token-level uncertainty. DDSUD dynamically evaluates both subsequence uncertainty and sample diversity, leveraging the representational power of BERT to select the most representative and effective samples, thereby improving model efficiency and accuracy. Particularly, DDSUD excels in scenarios with imbalanced labeled datasets, effectively addressing two major challenges in sentiment classification tasks: (1) the unreliable identification of informative samples caused by label imbalance, and (2) the redundancy in sample selection due to the lack of diversity awareness. By dynamically assessing both the uncertainty of subsequences and the diversity of representations, DDSUD selects diverse and valuable samples for annotation, significantly improving learning efficiency and model performance.

DDSUD integrates three complementary modules:

**Subsequence Uncertainty Estimation:** this module locates ambiguous segments within the input sequence using a head-tail pointer strategy and evaluates their uncertainty based on token-level representations obtained from BERT.

**Diversity-Aware Useful Sample Selection:** this module quantifies the semantic novelty of unlabeled samples by measuring the cosine distance between candidate samples and the labeled sample set, ensuring that the selected samples are both diverse and non-redundant.

**Combining Uncertainty and Diversity for Query Scoring:** the scores from the above two modules are fused through a dynamic weighting mechanism. A time-dependent parameter $\beta$ gradually shifts the focus from diversity to uncertainty across active learning iterations, allowing the query strategy to adapt as the labeled set grows.

As illustrated in Fig. 1, the DDSUD framework selects samples through dynamic uncertainty evaluation and diversity measurement. Subsequence uncertainty ($S_{\text{nor}}(X)$) is evaluated by calculating the cosine similarity of the [CLS] token embeddings from BERT, focusing on identifying uncertain subsequences. Sample diversity ($D_{\text{nor}}(X)$) is measured using pairwise similarity, ensuring that the selected samples span a wide feature space. The weighting parameter $\beta$ is dynamically adjusted, balancing uncertainty and diversity, and adapting as the labeled set increases.

Additionally, the head-tail segmentation mechanism introduced in Fig. 1 effectively groups and visualizes the distribution of samples, further ensuring the diversity of feature representations. This approach not only promotes the selection of diverse samples but also improves annotation efficiency, enhancing the model's ability to learn from diverse data.

These three modules work synergistically. In each iteration, DDSUD ranks all unlabeled samples based on their combined query scores and selects the top $h$ samples for annotation, thereby significantly improving model performance while minimizing labeling costs.

## Subsequence uncertainty estimation

In traditional active learning, sentence-level or token-level uncertainty is often used to measure the informativeness of a sample. However, these coarse-grained metrics fail to capture the localized ambiguities present in Chinese sentiment expressions. DDSUD addresses this limitation by estimating uncertainty at the subsequence level, enabling a more detailed analysis of sentence ambiguity.

Given an unlabeled sentence $X^u = \{x_1, x_2, \ldots, x_n\}$, we first encode it using BERT and apply a token-level classification layer to calculate the predicted class probabilities for each token:

$$p_{x_i,c} = \text{Softmax}(W \cdot h_{x_i} + b), \quad \forall x_i \in X^u, \ c \in C. \tag{1}$$

Here, $h_{x_i}$ represents the hidden representation of token $x_i$, $W$ and $b$ are the classifier parameters, and $C$ is the set of class labels.

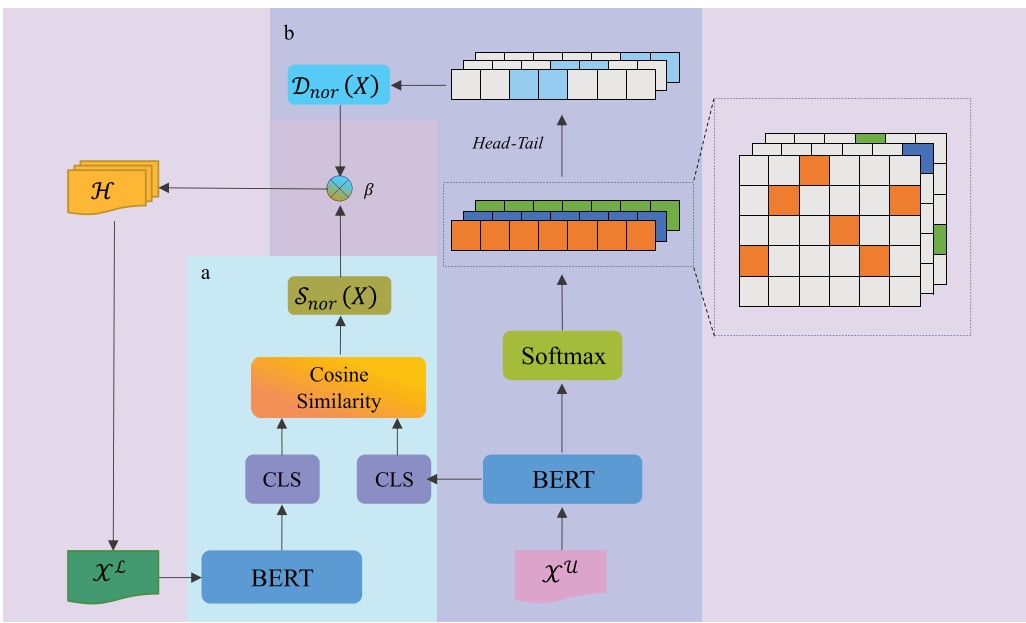

**Figure 1 DDSUD framework for sample selection based on uncertainty and diversity.**

To measure the uncertainty of each token, we use a least-confidence approach. The uncertainty score $\Phi_{x_i}$ for each token is calculated as follows:

$$\Phi_{x_i} = -\max_{c \in C} \log p_{x_i,c}. \tag{2}$$

To identify ambiguous subsequences within a sentence, we use a head–tail addressing mechanism. For each input $X^u$, we extract fixed-length subsequences from both the beginning and end (head and tail) of the sequence. For each candidate subsequence $subX$, we compute the average uncertainty score:

$$\Phi_{subX} = \frac{1}{|subX|} \sum_{x_i \in subX} \Phi_{x_i}. \tag{3}$$

The subsequence with the highest uncertainty score, $\Phi_{subX}$, is selected to represent the overall uncertainty for the instance:

$$S(X) = \max(\Phi_{head}, \Phi_{tail}). \tag{4}$$

To make the uncertainty scores comparable across instances, we normalize the raw uncertainty scores $S(X)$ using min–max normalization:

$$S_{nor}(X) = \frac{S(X) - \min(S)}{\max(S) - \min(S)}. \tag{5}$$

The normalized subsequence uncertainty score $S_{nor}(X)$ is then used in the query scoring process to prioritize ambiguous samples for annotation in active learning.

**Figure 2 Examples of partial labels for samples.**     

As shown in Fig. 2, this method is demonstrated with four example sentences. The model expresses high confidence in certain subsequences, such as those labeled with Degree and Trigger tags. For example, in sentences 2, 3, and 4, the pronoun "he" is interpreted as a Sentence Entity or Holder. However, in sentence 1, the conjunction "because," which signals causality, introduces more uncertainty in interpreting "he." This example illustrates how assessing the uncertainty in specific subsequences enables the model to precisely identify local areas of uncertainty.

Through this process, DDSUD offers a fine-grained approach for addressing sentiment ambiguities, enhancing the overall performance of sentiment analysis by targeting specific uncertainties in the data.

### Head–tail addressing algorithm

To identify the most uncertain regions within a sentence, we introduce the head-tail addressing strategy. This method extracts fixed-length subsequences from both the beginning and the end of the sentence, then evaluates the uncertainty of each subsequence. By focusing on the most uncertain parts, we avoid the need to evaluate all possible subsequences, improving both efficiency and precision in uncertainty localization, as illustrated in Fig. 3.

Given an unlabeled sequence $X^u = \{x_1, x_2, \ldots, x_n\}$ and a predefined subsequence length $m$, the following steps are executed:

1. **Extract Subsequences:**

   - Head subsequence: $subX_{\text{head}} = \{x_1, \ldots, x_m\}$
   - Tail subsequence: $subX_{\text{tail}} = \{x_{n-m+1}, \ldots, x_n\}$

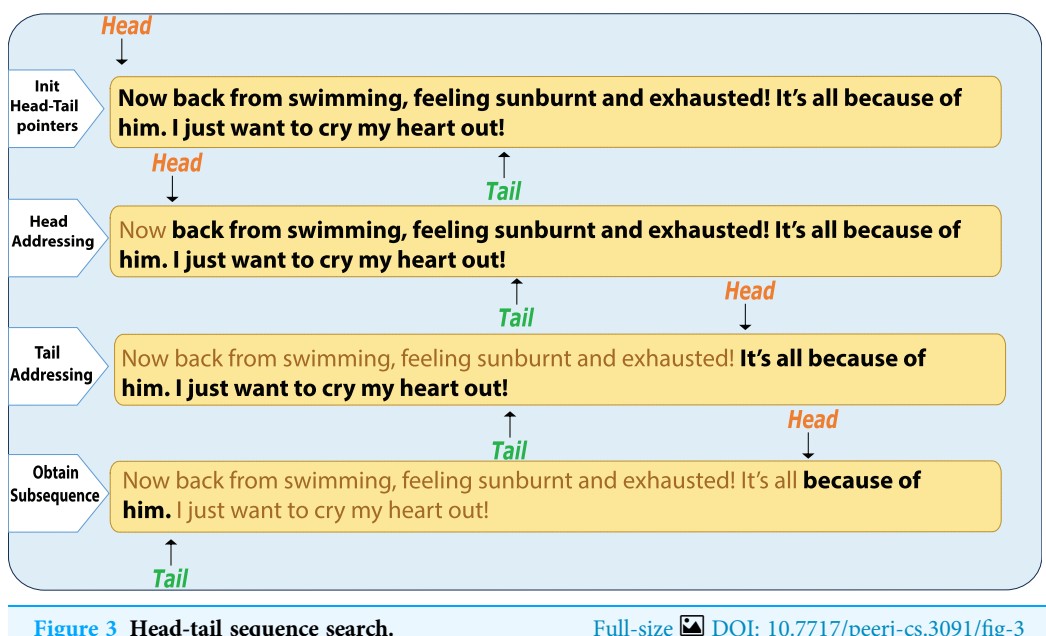

**Figure 3  Head-tail sequence search.**

2. **Compute Token-Level Uncertainty Scores:** for each token $x_i$ in the subsequences, we compute the uncertainty score using least-confidence scoring:

$$\Phi_{x_i} = -\max_{c \in C} \log p_{x_i,c}$$

where $p_{x_i,c}$ represents the model's predicted probability for token $x_i$ under class $c$.

3. **Compute Average Uncertainty Score for Each Subsequence:** the average uncertainty score for each subsequence is computed as:

$$\Phi_{\text{subX}} = \frac{1}{m} \sum_{x_i \in subX} \Phi_{x_i}.$$

4. **Assign the Final Uncertainty Score for $X^u$:** the uncertainty score for the entire sequence is the maximum of the two subsequences' uncertainty scores:

$$S(X^u) = \max \Phi_{\text{subX}_{\text{head}}}, \Phi_{\text{subX}_{\text{tail}}}.$$

This method allows the model to focus on the most uncertain regions of a sequence, improving both the efficiency and precision of uncertainty localization.

In each iteration of the querying process, the model $M$ is initially trained using a randomly selected sample set to calculate the overall uncertainty score, denoted as $\Phi_X$ for the entire sentence. This score is then stored as $\Phi$.

A pair of pointers, **head** and **tail**, are initialized to represent the beginning and end of the sentence, respectively. The head-tail addressing mechanism is then executed as follows:

1. **Head Pointer Movement:** The head pointer is moved one unit to the right, bringing the head and tail pointers closer together, creating a new subsequence

$subX = \{x_2, x_3, \ldots, x_n\}$. The uncertainty score $\Phi_{\text{subX}}$ for this subsequence is calculated. We then compare the magnitude of $\Phi$ with $\Phi_{\text{subX}}$:

- if $\Phi \geq \Phi_{\text{subX}}$, this indicates that the uncertainty score of the first token is relatively high, suggesting that the model is not very confident in predicting its label. In this case, the movement is reversed, and the head pointer addressing is terminated.
- If $\Phi < \Phi_{\text{subX}}$, it suggests that the uncertainty score of the first token is relatively low, indicating a higher confidence. The process continues by moving the head pointer to the next position.

2. **Tail Pointer Movement:** after handling the head pointer, the tail pointer is moved one unit to the left, forming a new subsequence $subX = \{x_i, x_{i+1}, \ldots, x_{n-1}\}$. The uncertainty score $\Phi_{\text{subX}}$ is recalculated. Similarly to the head pointer step, if the new uncertainty score is higher than the current score, the movement is reversed, terminating the tail pointer addressing.

By performing these head-tail addressing operations, the subsequence with the highest uncertainty score is identified, which corresponds to the most uncertain region within the sequence.

## Diversity-aware informative sampling

In active learning, selecting samples solely based on uncertainty may lead to redundant instances that provide similar information to what the model has already seen. To mitigate this, DDSUD incorporates a diversity-based strategy that encourages the selection of semantically novel and informative samples.

Let $X^L$ be the labeled set and $X^u$ an unlabeled candidate. We use the hidden representation of the [CLS] token from BERT as the semantic embedding of a sequence, denoted as $h_X \in \mathbb{R}^d$. The diversity of $X^u$ is assessed by its dissimilarity to the labeled set embeddings:

$$\text{sim}(X^u, X_i^L) = \frac{h_{X^u} \cdot h_{X_i^L}}{||h_{X^u}|| \cdot ||h_{X_i^L}||}, \quad \forall X_i^L \in X^L. \tag{6}$$

We then compute the average similarity between $X^u$ and all labeled samples:

$$\bar{S}(X^u) = \frac{1}{|X^L|} \sum_{X_i^L \in X^L} \text{sim}(X^u, X_i^L). \tag{7}$$

The diversity score $D(X^u)$ is defined as the inverse of this average similarity:

$$D(X^u) = 1 - \bar{S}(X^u). \tag{8}$$

To normalize the diversity scores across the unlabeled pool, we apply min–max normalization:

$$D_{\text{nor}}(X^u) = \frac{D(X^u) - \min(D)}{\max(D) - \min(D)}. \tag{9}$$

1. Extract the [CLS] representation $h_{X^u}$ for each unlabeled instance using BERT.
2. Compute cosine similarity between $h_{X^u}$ and each $h_{X_i^L}$ in the labeled set.
3. Compute the average similarity $\bar{S}(X^u)$ and derive diversity score $D(X^u) = 1 - \bar{S}(X^u)$.
4. Apply normalization to get $D_{\text{nor}}(X^u)$ for scoring.

This strategy ensures that selected samples not only contain uncertain information but also provide novel semantic content, thereby enhancing the diversity of the training set and improving generalization.

## Combining uncertainty and diversity for query scoring

To effectively guide sample selection during active learning, DDSUD combines two complementary scores for each unlabeled instance: subsequence-level uncertainty and semantic diversity. Each component captures a different aspect of informativeness:

- $S_{\text{nor}}(X)$: the normalized uncertainty score derived from subsequence entropy.
- $D_{\text{nor}}(X)$: the normalized diversity score computed from cosine similarity with labeled samples.

To balance these two signals dynamically during training, we define a time-dependent scoring function:

$$Q(X) = \beta \cdot D_{\text{nor}}(X) + (1 - \beta) \cdot S_{\text{nor}}(X). \tag{10}$$

Here, $\beta \in [0, 1]$ is a weighting factor that controls the emphasis between diversity and uncertainty. Following a curriculum-style schedule, we set $\beta$ to gradually decay with respect to the active learning round $t$:

$$\beta_t = \beta_0 \cdot \gamma^t \tag{11}$$

where $\beta_0$ is the initial weight (*e.g.*, 0.9) and $\gamma \in (0, 1)$ is a decay coefficient.

This mechanism encourages exploration in the early stages (favoring diverse samples), and gradually shifts focus toward uncertainty-based exploitation as more labeled data become available.

At each iteration, all unlabeled instances are scored using $Q(X)$, and the top-$h$ ranked samples are selected for annotation.

## Step-by-step query workflow

In DDSUD, the three core modules operate in a coordinated sequence to perform query selection during each active learning iteration. The process is executed as follows:

1. For each unlabeled sample $X^u$, apply the head–tail addressing strategy to locate its most ambiguous subsequence, and compute the normalized uncertainty score $S_{\text{nor}}(X^u)$.
2. Compute the diversity score $D_{\text{nor}}(X^u)$ by evaluating its semantic distance to the labeled set based on [CLS] embeddings.

3. Combine both scores using the dynamic weighting parameter $\beta_t$ to calculate the final query score:

$$Q(X^u) = \beta_t \cdot D_{\text{nor}}(X^u) + (1 - \beta_t) \cdot S_{\text{nor}}(X^u).$$

4. Rank all unlabeled samples according to $Q(X^u)$ and select the top-$h$ samples for annotation.

5. Add the newly labeled samples to the training set, and continue to the next round.

This workflow ensures that DDSUD adaptively selects samples that are not only uncertain but also semantically novel, maximizing annotation efficiency across iterations.

# EXPERIMENTS AND RESULTS ANALYSIS

## Dataset

The dataset used in this study is the publicly available Chinese Textual Affective Structure (CTAS) dataset, specifically designed for affective structure recognition in Chinese social media text. Each sentence in CTAS is annotated at the token level using the BIO tagging scheme, covering emotion-related components such as Trigger, Degree, Cause, and Holder, among others. In addition to span-level role labels, each token is also assigned a part-of-speech tag and semantic category, enabling fine-grained analysis of affective structures within complex linguistic contexts.

To ensure the robustness and generalizability of experimental results, the dataset is partitioned into training, validation, and test subsets using an 8:1:1 ratio. Furthermore, five-fold cross-validation is adopted: in each fold, data is randomly split, the model is trained on distinct partitions, and the final results are averaged across all folds. This design helps mitigate variability caused by specific data splits and provides a more stable performance estimate.

The CTAS dataset is publicly accessible online (https://github.com/pdsxsf/CTAS).

## Baseline strategies

To evaluate the efficacy of our method, we conducted a comparative analysis against established baseline approaches. A detailed description of these methods is provided below.

Let $X$ denote the set of unlabeled samples, where $X_U$ represents the unlabeled dataset and $X_L$ represents the labeled dataset. In this framework, $X$ specifically refers to individual samples drawn from the unlabeled dataset $X_U$, and $S_{nor}(X)$ represents the diversity score calculated based on the similarity between each sample and the labeled dataset $X_L$. This method allows for effective quantification of the representativeness of unlabeled samples, enabling the prioritization of samples with higher information content during the sampling process.

**DDSUD:** our approach employs a BERT-based framework that enhances dataset diversity through subsequence uncertainty detection, diversity-driven sample selection, and dynamic weighting. These mechanisms allow DDSUD to adaptively balance uncertainty

and diversity throughout the active learning process, thereby improving both efficiency and performance, especially when dealing with imbalanced datasets.

**Token Entropy (TE):** Token Entropy is a sample selection strategy that quantifies the uncertainty of each token within a given sample. The model first predicts the label probability distribution for each token, then calculates the information entropy for each token, selecting those with the highest entropy for labeling.

**Entity-Aware Subsequence Active Learning (EASAL):** this method (*Liu et al., 2023*) leverages BERT and queries entity-aware subsequences for each sentence. The uncertainty scores of these subsequences are then ranked in descending order to select the highest-ranking subsequences, thereby maximizing labeling information with a limited number of labels.

**Least Confidence (LC):** the Least Confidence (LC) method (*Lewis, 1995*) selects data points with the lowest confidence scores among the most probable label sequences, aiming to annotate the most uncertain samples. In our experiments, the confidence is computed as the probability of the most likely label sequence predicted by the BERT+CRF model, and samples with the lowest scores are selected in each iteration. However, this method may inadvertently prioritize longer or syntactically complex sequences, potentially overlooking other informative samples during training.

**RANDOM:** in addition to the previously mentioned baselines, RANDOM was selected as an additional baseline strategy.

## Environment and parameter settings

The computational environment and experimental parameters used in this work are summarized in Table 1. All experiments were performed on a Linux-based system utilizing an NVIDIA GeForce RTX 3090 GPU, ensuring robust computational capacity for training and evaluating the BERT+CRF framework. Critical hyperparameters such as learning rate and batch size were systematically optimized to balance model accuracy with resource efficiency.

In particular, the parameter initial selection fraction refers to the proportion of the unlabeled dataset that is randomly selected for initial annotation before the active learning iterations begin. This initial seed set provides labeled samples to bootstrap model training. For example, a value of 5% indicates that 5% of the unlabeled data is labeled in advance prior to querying.

Given BERT's training stability and structural suitability for sequence labeling tasks, it is adopted as the backbone encoder in this study. Compared with models such as RoBERTa and ALBERT, BERT is easier to fine-tune under low-resource settings, offers more efficient inference, and has been widely applied in active learning for both sequence labeling and text classification tasks. To avoid potential confounding effects introduced by architectural differences during the evaluation of sampling strategies, we prioritize a controllable and reproducible standard model.

Table 1 Experimental environment and parameters.

| Environmental parameters | Value |
| --- | --- |
| Operating system | Linux |
| GPU | NVIDIA GeForce RTX 3090 |
| Model | BERT+CRF |
| Learning rate | 5e−5 |
| Max len | 256 |
| Epoch | 100 |
| Batch size | 64 |
| Query batch fraction | 5% |
| Initial selection fraction | 5% |
| Length of subsequence | 2 |

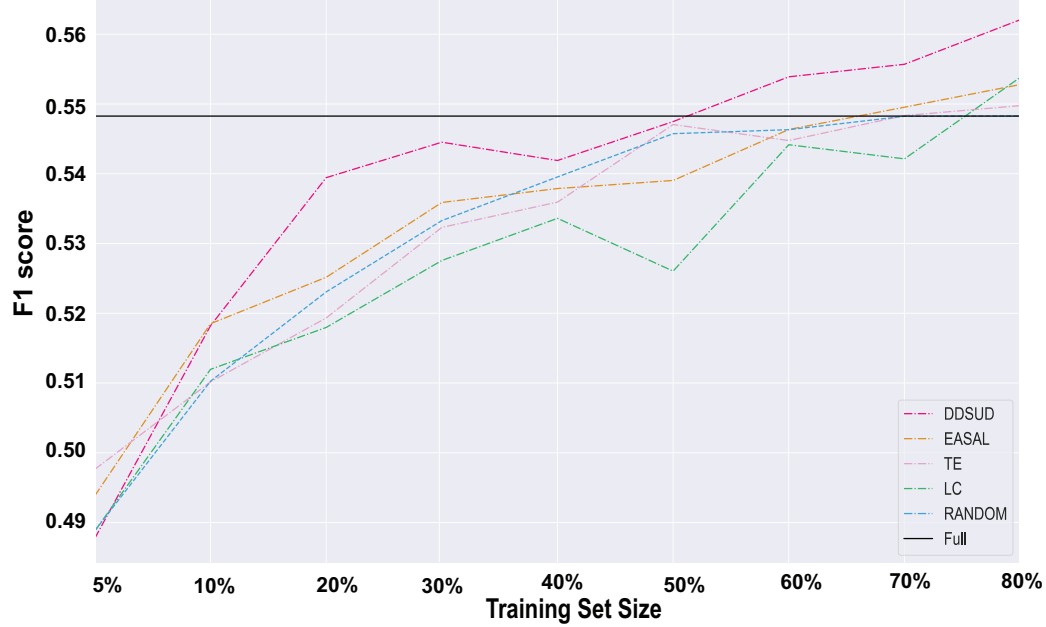

Figure 4 Comparative strategy effectiveness.     

## Comparative experiments

To assess the effectiveness of DDSUD, we conducted comparative experiments against several baseline strategies, and the results are shown in Fig. 4.

As shown in Fig. 4, DDSUD consistently outperforms other active learning methods across various training set sizes. The F1-score of DDSUD improves steadily as training data increases, with particularly large gains under low-resource settings. At 50% labeled data, DDSUD achieves an F1-score of 0.5430, closely approaching the 0.5479 score of a fully supervised BERT+CRF model. The performance gap is less than 0.5%, demonstrating that DDSUD can reach near full-supervision accuracy with only half of the labeling cost.

**Table 2 Minority class label variation.**

| Label | Sampling strategy | Initial quantity | Quantity after first iteration | Quantity after second iteration |
|---|---|---|---|---|
| Cause | DDSUD | 50 | 91 | 129 |
| | SUD | 50 | 74 | 110 |
| Negation | DDSUD | 19 | 29 | 54 |
| | SUD | 19 | 28 | 46 |
| Property | DDSUD | 6 | 23 | 32 |
| | SUD | 6 | 12 | 20 |
| Compared entity | DDSUD | 4 | 8 | 11 |
| | SUD | 4 | 7 | 13 |

These findings highlight DDSUD's capacity to effectively combine diversity and subsequence-level uncertainty for sample selection. In contrast, EASAL—based solely on subsequence uncertainty—shows only moderate improvements. Although EASAL performs better than random sampling in certain settings, its overall gains remain limited compared to DDSUD, indicating that uncertainty alone is insufficient.

TE and LC strategies exhibit relatively stable but less impressive performance, particularly under smaller training sizes. This suggests that focusing solely on confidence or entropy may not be effective for complex sequence labeling tasks.

Overall, DDSUD provides a more comprehensive sample selection mechanism by dynamically integrating uncertainty and diversity, making it especially suitable for class-imbalanced, low-resource scenarios. Additional evidence of robustness is provided in Table 2, which shows how DDSUD better expands the coverage of minority labels over multiple selection rounds.

## Ablation study

To clarify the contributions of subsequence uncertainty detection and the dynamic weighting factor in the DDSUD framework, we designed two simplified model variants and conducted systematic comparisons against the full DDSUD model. The details are as follows:

**Model Variants.**

- **DDSUD (Full Model):** incorporates both subsequence uncertainty detection and a dynamic weighting factor $\beta$, which adaptively balances uncertainty and diversity during sample selection.
- **DDUD (w/o Subsequence Uncertainty):** removes subsequence-level uncertainty scoring while retaining the dynamic weighting strategy, relying solely on overall sample-level uncertainty for selection. This variant is used to isolate the contribution of subsequence analysis.
- **SUD (Static Weight):** replaces dynamic $\beta$ with a fixed value ($\beta = 0.5$), while keeping the subsequence uncertainty mechanism intact. This allows us to assess the effect of adaptive weighting on learning flexibility and performance.

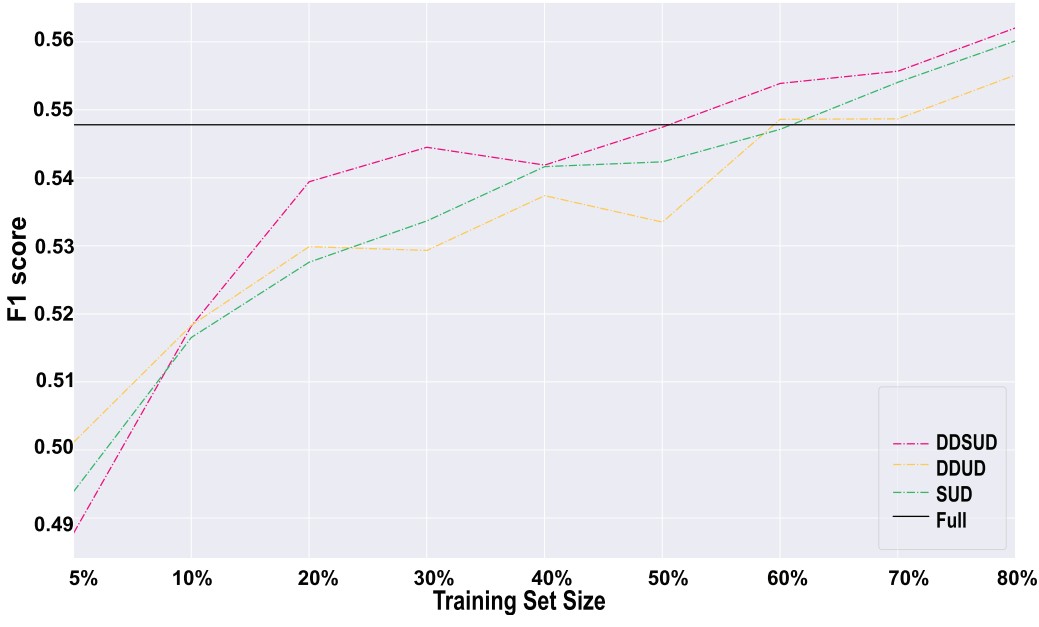

**Figure 5  Results of ablation experiments.**

**Experimental Setup and Failure Criteria.** All models are trained on the same dataset under consistent settings (BERT+CRF, learning rate $5 \times 10^{-5}$, batch size 64, maximum sequence length 256, and 5-fold cross-validation) to ensure a fair comparison. We define a failure round as one in which the F1-score falls below 0.50 or when the sampling strategy leads to non-convergent training.

**Comparison Results.** As shown in Fig. 5, the results consistently demonstrate the effectiveness of both core modules. Across training set proportions from 20% to 80%, the DDSUD model consistently outperforms both simplified variants in F1-score. This confirms the complementary benefits of combining fine-grained subsequence uncertainty with a dynamic sample weighting strategy. The subsequence mechanism provides more precise identification of ambiguous regions, while dynamic adjustment improves the model's adaptability to changing data distributions.

**Conclusion.** The ablation study indicates that the two core modules in DDSUD offer mutual reinforcement. Subsequence-level uncertainty improves the focus of the model on critical segments, and dynamic weighting enhances the flexibility of the sample selection process. Together, they contribute to superior performance and better generalization under low-label, imbalanced scenarios.

## Case analysis

The effectiveness of dynamic probing is evaluated by analyzing the distribution of labels assigned to minority class samples in the training dataset. Table 2 presents a statistical comparison between DDSUD and its variant SUD, emphasizing the improvements introduced through dynamic detection.

When utilizing the DDSUD sampling strategy, the number of samples for each minority class label increased substantially after the first and second iterations. For instance, the Cause label initially had 10 samples, which increased to 91 after the first iteration and further to 129 after the second iteration. Conversely, the Subsequence Uncertainty Detection (SUD) sampling strategy resulted in smaller increases, with the same label reaching 74 samples after the first iteration and 110 after the second.

This observation indicates that the DDSUD sampling strategy surpasses both the SUD sampling strategy and baseline approaches in increasing the number of minority class samples. By dynamically adjusting the weights of subsequence uncertainty and diversity, the DDSUD strategy effectively augments minority class samples in datasets with imbalanced label distributions, enhancing the learning capacity of the model for minority classes and improving performance on imbalanced datasets.

## CONCLUSION

In this work, we propose DDSUD, a novel BERT-based active learning framework designed to tackle challenges in sentiment classification for Chinese text, particularly in scenarios with imbalanced datasets and limited labeled data. DDSUD integrates subsequence uncertainty detection, diversity-driven sample selection, and dynamic weighting, offering a comprehensive and adaptive approach to active learning. This framework significantly improves sentiment analysis performance, especially in resource-constrained and challenging environments.

Experimental results show that DDSUD outperforms traditional active learning methods, particularly in improving the representation of underrepresented categories such as "Cause" and "Negation". By focusing on granular subsequences and utilizing dynamic probing, DDSUD facilitates effective sample selection, leading to significant gains in model accuracy and generalization. Ablation studies further emphasize the key roles of subsequence uncertainty detection and dynamic weighting. Together, these components enable DDSUD to adapt to evolving data distributions and precisely target task-specific information.

Looking ahead, we outline several strategic research directions: (1) Structural uncertainty extension: applying the core principles of DDSUD to other Chinese NLP tasks involving complex structure prediction, such as event extraction and semantic role labeling; (2) Data expansion: collaborative annotation of additional datasets for sentiment structure recognition when resources permit; (3) Generalization potential: adapting structural uncertainty quantification to broader sequence labeling tasks with comprehensive state of the art (SOTA) comparisons; (4) Architecture refinement: optimizing the dynamic weighting mechanism through data-driven techniques for enhanced contextual adaptability; (5) Real-time integration: exploring reinforcement learning and stream processing implementations for rapid-insight applications like social media monitoring.

Through these developments, the innovative design of DDSUD positions it as a foundational framework for advancing active learning techniques, particularly in resource-constrained and linguistically complex scenarios. Its scalability and adaptability

offer transformative potential for sentiment analysis and beyond, paving the way for more equitable and efficient learning systems.

### Funding

This work was supported by the MOE (Ministry of Education of China) Project of Humanities and Social Sciences (No. 24YJAZH149), the Henan Province key research and development project (No. 231111211300) and the Henan Province Science and Technology Development Plan Project (No. 242102521027). There was no additional external funding received for this study. The funders had no role in study design, data collection and analysis, decision to publish, or preparation of the manuscript.

### Grant Disclosures

The following grant information was disclosed by the authors:
MOE (Ministry of Education of China) Project of Humanities and Social Sciences: 24YJAZH149.
Henan Province key research and development project: 231111211300.
Henan Province Science and Technology Development Plan Project: 242102521027.

### Competing Interests

The authors declare that they have no competing interests.

### Author Contributions

- Shufeng Xiong conceived and designed the experiments, authored or reviewed drafts of the article, and approved the final draft.
- Yibo Si performed the experiments, analyzed the data, performed the computation work, prepared figures and/or tables, and approved the final draft.
- Guipei Zhang performed the experiments, analyzed the data, performed the computation work, prepared figures and/or tables, and approved the final draft.
- Bingkun Wang conceived and designed the experiments, authored or reviewed drafts of the article, and approved the final draft.
- Guang Zheng analyzed the data, prepared figures and/or tables, authored or reviewed drafts of the article, and approved the final draft.
- Haiping Si conceived and designed the experiments, authored or reviewed drafts of the article, and approved the final draft.

### Data Availability

The data and code are available at Zenodo: henau-nlp. (2025). K49Z/DDSUD: public release of DDSUD (v1.0). Zenodo. https://doi.org/10.5281/zenodo.15614305.

## Supplemental Information

Supplemental information for this article can be found online at http://dx.doi.org/10.7717/peerj-cs.3091#supplemental-information.

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
