# Peer review of "DDSUD: dynamically detecting subsequence uncertainty and diversity for active learning in imbalanced Chinese sentiment analysis"

_PeerJ Computer Science, doi:10.7717/peerj-cs.3091_

## Round 0.1 · original submission · Major Revisions

Reviewer 1 ·

Basic reporting

The paper proposes the DDSUD framework, which is meant to enhance active learning for imbalanced Chinese sentiment analysis by addressing subsequence uncertainty and improving dataset diversity.

The paper is within the scope of the journal. Also, the topic is currently of main interest to the NLP (Natural Language Processing) scientific community. The article is correctly organized, but the content of each section is superficial (i.e., important details are missing).

The introduction should more explicitly state and briefly discuss the contributions of the paper to be more broadly understood. Moreover, the structure/organization of the paper needs to be specified at the end of the Introduction section. Section 2 should enhance the understanding of the state of the art in the field by: a) including more recent papers in the survey (NLP is a rapidly advancing field, and, from this perspective, citing a lot of papers that are 5-20 years old is strange); and, b) precisely identify the research gap this paper covers in the context of existing works (it is not sufficient to only briefly describe some papers, without analyzing their outputs in the context of the proposed framework); Section 3, devoted to the DDSUD framework presentation, is confusingly organized, without giving all the details for the framework to be reproducible.

Experimental design

The experimental part of the paper needs to be better designed and also better structured. The dataset (subsection 4.1) employed in the framework evaluation is not available to the reader (i.e., no reference or a link to a repository is specified), and the internal structure of the original dataset is not presented. To evaluate the framework's potential, it needs to be compared with other state-of-the-art frameworks and not with general (e.g., LC or TE) strategies. Also, the ablation study is confusing, since its methodology is not clearly presented.

The results, as displayed in Figure 4, are not impressive, as DDSUD is only marginally ahead of EASAL (please note that the reference for EASAL, namely Liu et al., 2022, is wrong, so there are no details on how the comparison has been done).

Validity of the findings

The proposed framework is validated using only one dataset, with no solid comparison to other state-of-the-art methods. Because of these, additional and extensive validations must be carried out to be sure the DDSUD framework is a step forward in enhancing sentiment analysis in Chinese.

Additional comments

- The reason to choose BERT and not another ML method, including BERT-like models (e.g., RoBERTa, DistilBERT, ALBERT, etc.), is not presented;
- Abstract, lines 21-22: It is written that “Experimental results confirm that the DDSUD method achieves performance comparable to fully supervised training schemes with only 50% of the data labeled in imbalanced datasets,” but this claim is not backed up by proper reasons and experimental evidence.
- The specificity of sentiment analysis in Chinese should be better explained. Why do general models (e.g., in English) not meet the expectations?
- Some details about how LC….
- Line 301: It is written that “The weight factor beta dynamically changes throughout the active learning iterations,” but no strategy/equation is provided to sustain this.
- Table 1: X is not known at the first time step 2 is performed;
- Table 2: What “Ori choose fraction” represent?
- Line 526: There is a typo (OZKURT instead of Ozkurt).
- Line 540: the reference Settles, B. (2009). It is not fully specified;

Reviewer 2 ·

Basic reporting

All comments have been added in detail to the last section.

Experimental design

All comments have been added in detail to the last section.

Validity of the findings

All comments have been added in detail to the last section.

Additional comments

Review Report for PeerJ Computer Science
(DDSUD: Dynamically detecting subsequence uncertainty and diversity for active learning in imbalanced Chinese sentiment analysis)

1. In this study, a BERT-based active learning framework called DDSUD, which considers subsequence uncertainty and diversity, is proposed; the method achieves performance comparable to fully supervised approaches using only partially labeled data and outperforms existing active learning methods.

2. In the Introduction, sentiment classification, the use of deep learning and machine learning approaches in this regard, and the importance of the subject are explained at a basic level. In order to emphasize the study even more at the end of this section, the differences of this study from the literature, its contributions to the literature and its originality should be stated in detail.

3. In the Related work section, sequence sabeling with active learning, sample information, and sequence labeling methods in deep learning are sufficiently and explanatorily mentioned in relation to the literature.

4. When the BERT-based framework proposed within the scope of the study is examined, it is observed that it has a certain level of originality. However, in this section, it should be stated in more detail why this transformer-based model is preferred, especially despite the many different deep learning-based models that can be used in the literature.

5. The type and amount of dataset used in the study is appropriate for the study topic. However, it should be explained in detail why the dataset distribution is divided into 80% training, 10% validation and 10% testing. Using cross validation is very positive in terms of the quality of the results.

6. Although the parameter selections and types seem appropriate when examined in detail, how the values of the parameters are determined for this part should be detailed.

In conclusion, the study is interesting and high quality, but all the above parts should be paid close attention.

---

## Round 0.2 · accepted · Accept

The authors correctly addressed the requests of the reviewers and therefore I can recommend this article for acceptance.

Reviewer 1 ·

Basic reporting

All criteria regarding "basic reporting" issues have been met. The authors have successfully addressed all my previous comments and concerns.

Experimental design

All criteria regarding the experimental validation have been met. The authors have successfully addressed all my previous comments and concerns.

Validity of the findings

All criteria regarding the "validity of the findings" have been met. The authors have successfully addressed all my previous comments and concerns.

Reviewer 2 ·

Basic reporting

-

Experimental design

-

Validity of the findings

-

Additional comments

Thanks for the revision, both the responses to the reviewer comments and the changes made to the paper are sufficient.